# Developing Nanodisc-ID for label-free characterizations of membrane proteins

Huan Bao [1 ✉]

Membrane proteins (MPs) influence all aspects of life, such as tumorigenesis, immune response, and neural transmission. However, characterization of MPs is challenging, as it often needs highly specialized techniques inaccessible to many labs. We herein introduce nanodisc-ID that enables quantitative analysis of membrane proteins using a gel electrophoresis readout. By leveraging the power of nanodiscs and proximity labeling, nanodisc-ID serves both as scaffolds for encasing biochemical reactions and as sensitive reagents for detecting membrane protein-lipid and protein-protein interactions. We demonstrate this label-free and low-cost tool by characterizing a wide range of integral and peripheral membrane proteins from prokaryotes and eukaryotes.

[1] Department of Molecular Medicine, The Scripps Research Institute, Jupiter, FL, USA. ✉email: hbao@scripps.edu

Signal transduction across cellular membranes is crucial for life and often involves the action of membrane proteins (MPs)[1,2]. The importance of these membrane-embedded molecules is underscored by the fact that they constitute over 60% of drug targets for numerous human diseases[3]. Due to the hydrophobic nature of lipids, it is, however, technically challenging to obtain mechanistic understandings of MPs. In particular, many low-affinity membrane protein–lipid and protein–protein interactions remained untapped[2]. Hence, a pressing need exists to develop simple and straightforward methods that can rapidly interrogate these interactions. To tackle this challenge, we set out to marry nanodiscs with proximity labeling (PL) for facile and sensitive characterizations of MPs.

Over the past few years, PL has emerged as a powerful approach for mapping protein–protein, protein–RNA, and protein–DNA interactomes[4]. Genetic fusion of PL enzymes (e.g., APEX and BioID) to the protein of interest could readily label interaction partners within a distance of 10–20 nm. This approach provides unmet sensitivities to detect transient interactions, thereby revealing critical mediators involved in multiple signaling processes[5,6]. We are inspired by these studies and posit that PL would be a great tool to study protein–membrane interactions if these labeling enzymes could be restrained in a 10–20 nm lipid bilayer.

On this front, nanodisc (ND) is an ideal system to accommodate PL enzymes[7]. Developed by the Sligar laboratory, NDs enclose nanoscale lipid bilayers via amphipathic membrane scaffold proteins (MSPs)[8], endowing otherwise insoluble lipids and MPs water-soluble and remarkably stable in solution[9]. Therefore, NDs have profoundly advanced biophysical and structural studies of MPs in the past decade[10]. Here, we demonstrated that NDs could also serve as an excellent platform for the deployment of PL enzymes to characterize MPs (Fig. 1A). We described our effort in developing the assembly of nanodiscs with PL enzymes to identify membrane interactions (nanodisc-ID). Moreover, we showcased the utility of nanodics-ID for profiling membrane protein–lipid and protein–protein interactions involved in a multitude of prokaryotic and eukaryotic signaling pathways. Together, nanodisc-ID could serve as a powerful and versatile approach for biochemical dissections of MPs.

## Results

### Identification of compatible PL enzymes for conjugation with NDs.
First, we screened a panel of PL enzymes to test their compatibility with NDs for detecting the interaction between peripheral membrane proteins (pMPs) and lipids. To do so, we anchored PL enzymes onto MSP1D1 encircled nanodiscs through the interaction of the His-tag on these enzymes with $Ni^{2+}$-NTA functionalized lipids (Fig. 1A), and then assayed if they could label a PS binding protein, synaptotagmin-1 (syt1)[11], in a lipid dependent manner (Fig. 1B). We tested three classes of proximity labeling enzymes: (1) APEX2 derived from peroxidase[12,13]; (2) BioID and TurboID derived from biotin ligase[14,15]; (3) PafA, a bacterial ubiquitin-like protein ligase[16]. APEX2, BioID, and TurboID will tag substrate with biotin, which could be readily detected by electrophoresis upon incubation with streptavidin (SA). On the other hand, PafA could ligate a small protein Pup onto nearby targets. Thus, labeling of syt1 by NDs harboring PL enzymes could be readily detected by a shift in mobility on SDS-PAGE (Fig. 2F), of which we have observed using APEX2, BioID, and TurboID (Fig. 1B). In this assay, APEX2 and TurboID exhibited higher labeling efficiency (Fig. 1B) and thus were used for developing the nanodisc-ID approach. As APEX2-mediated PL reactions require $H_2O_2$ that oxidizes lipids, we thereafter only used it for detecting membrane protein–protein interactions, whereas focused on applying TurboID for protein–lipid interactions.

### Developing Nanodisc-ID for protein–lipid interactions.
Despite the robust efficiency in detecting syt1-PS interaction, we also observed moderate levels of labeled syt1 by PL enzymes using NDs containing only PC lipids (Fig. 1B). This result is surprising as syt1 binds PC much weaker than PS[17]. Furthermore, the lipid specificity of syt1 was not captured (Fig. 1B and D), and other low-affinity protein–lipid interactions could not be detected (Fig. 1C). We suspected that these data were due to the use of $Ni^{2+}$-NTA conjugated lipids; proteins such as syt1 could bind cationic ions (e.g., $Ni^{2+}$) and the lipid anchored TurboID could constrain the surface area of NDs for pMP association. Nevertheless, we found that the labeling efficiency of syt1 is correlated with the percentage of $Ni^{2+}$-NTA conjugated lipids, suggesting that PL labeling efficiencies reflect the affinities of protein–lipid interactions (Fig. 1B and D).

To bypass the need for the use of $Ni^{2+}$-NTA conjugated lipids, we posited that the genetic fusion of MSP with TurboID could provide an ideal solution for both of the problems (Fig. 2A). Unfortunately, the fused protein, TurboID-MSP1D1, was mostly insoluble and exhibited a 10-fold decreased labeling efficiency (Fig. 2B). We thus screened an array of MSPs (Fig. 2B)[18–21]. Gratifyingly, fusion of TurboID with a membrane scaffold peptide (MSP-18A), yielding TurboID-18A, could still form homogenous NDs with a diameter of ~10–15 nm (Fig. 2B-D and Fig. S1), as characterized using size-exclusion chromatography (SEC) and negative stain EM. More importantly, MSP-18A did not perturb the activity of TurboID for detecting the interaction between syt1 and lipids (Fig. 2B and F), and TurboID-18A NDs were stable on ice for at least 2 days (Fig. S2). In addition, the labeling efficiencies of syt1 by TurboID-18A NDs were consistent with its lipid specificity (Figs 2F, 3A and supplementary data 1). As such, the highest labeling of syt1 occurred in the presence of PI lipids, and syt1 labeling by TurboID-18A NDs containing PS lipids was inhibited by regular MSP1D1 nanodiscs containing PI, but not PC lipids (Fig. S3). These data suggested that the nanodisc-ID approach could determine the specificity of protein–membrane interactions in vitro, even though proximity labeling tends to capture unspecific and transient binding events in vivo.

Moreover, NDs encased by TurboID-18A were able to label weak pMP binding to lipids (Fig. 3; cpx2 and EIIA$^{Glc}$). Using TurboID-18A NDs prepared with different lipids, we could readily profile the lipid-binding specificity of several pMPs (Fig. 3A, Fig. S1, S4, and supplementary data 1,). In control experiments, GST (Glutathione-S-transferase) that does not interact with lipids, was not labeled by TurboID-18A NDs at all conditions (Fig. 3, Fig. S4, and supplementary data 1).

Interestingly, we noticed that syt1 and SecA were much better biotinylated by TurboID-18A NDs than cpx and EIIA$^{Glc}$, indicating that the labeling efficiencies were dependent on the affinities of pMPs for lipids (Fig. 3A). Thus, we asked if equilibrium titrations of nanodisc-ID could be used to determine such affinities. To test this possibility, we utilized nanodisc-ID to characterize our collection of pMPs (syt1, SecA, cpx2, EIIA$^{Glc}$), whose lipid-binding affinities range from 10 nM to 100 μM[22–25]. These results revealed that all of these lipid-specific interactions could be detected using nanodisc-ID, and equilibrium titrations revealed that the apparent binding affinities were consistent with previous studies (Fig. 3B, supplementary data 1).

### Developing Nanodisc-ID for membrane protein–protein interactions.
Next, we tested if nanodisc-ID could be used to study membrane protein–protein interactions. On this front, we showcased the power of nanodisc-ID using the maltose ATP-binding cassette (ABC) transporter MalFGK$_2$ from E. coli and the SNARE complex that mediates vesicle exocytosis in neurons.

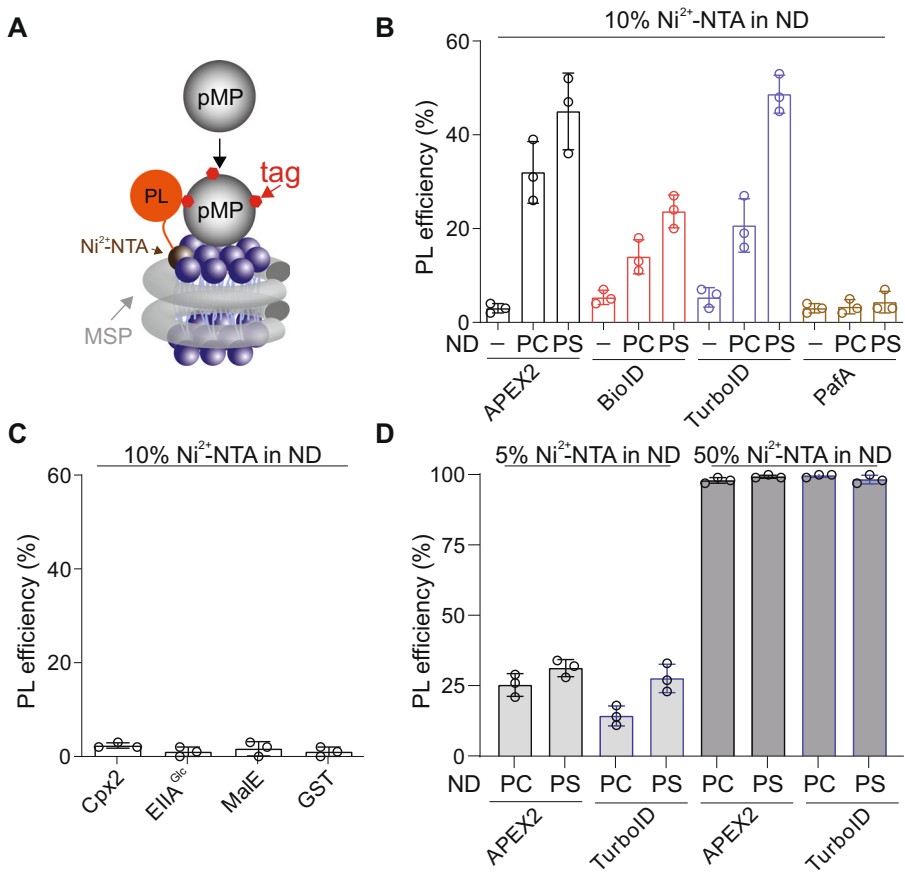

**Fig. 1 Screening of PL enzymes suitable for developing nanodisc-ID. A** In the initial screening, PL enzymes were attached to nanodiscs through the interaction of their N-terminal His-tag with $Ni^{2+}$ containing lipid bilayers. **B** Quantification of syt1 labeling efficiency by the indicated PL enzymes immobilized on the lipid bilayer of PC and PS nanodiscs containing 10% $Ni^{2+}$-NTA functionalized lipids. **C** Labeling efficiencies of low-affinity lipid-binding proteins, cpx2 and $EIIA^{Glc}$, by TurboID immobilized on the lipid bilayer of nanodiscs. Maltose-binding protein (MalE) and Glutathione-S-transferase (GST) were used as controls. **D** Quantification of syt1 labeling efficiency by the indicated PL enzymes immobilized on the lipid bilayer of PC and PS nanodiscs containing 5% or 50% $Ni^{2+}$-NTA functionalized lipids. Data are shown as mean ± s.d., $n = 3$ independent experiments.

$MalFGK_2$ alternates between the inward- and outward-facing configuration to translocate substrates across membrane using the energy of ATP hydrolysis[26]. This transport mechanism requires the association of the maltose-binding protein (MalE). MalE binding depends on the conformational state of $MalFGK_2$; the affinity of MalE to the inward- and outward-facing transporter is 80 nM and 5 µM, respectively[27]. Since PL by TurboID needs ATP hydrolysis that also controls the conformational equilibrium of $MalFGK_2$, we turned to APEX2 for developing the nanodisc-ID approach that suitable for detecting the interaction between MalE and $MalFGK_2$. We fused APEX2 with MSP1D1, and the resulting APEX2-MSP1D1 was able to form homogenous and stable NDs with a diameter of ~10–12 nm, as shown by SEC and negative stain EM (Fig. 4 and S2). Moreover, we could functionally reconstitute $MalFGK_2$ in APEX2-MSP1D1 NDs (Fig. S1 and S5), in which the ATPase activity of $MalFGK_2$ was responding to the addition of MalE and maltose. We then used APEX2-MSP1D1 NDs to quantify the interaction of MalE with $MalFGK_2$ in both the inward- and outward-facing states (Fig. 5A). Upon incubation with $MalFGK_2$ reconstituted in APEX2-MSP1D1 NDs, labeling of MalE is readily observed and further enhanced with a ~50-fold increase in apparent binding affinity by the addition of nonhydrolyzable nucleotides that induced the outward-facing conformation.

Furthermore, APEX2-MSP1D1 NDs could be used to study the impact of regulators on the SNARE proteins that mediate most intracellular membrane fusion events[28]. The pairing of the vesicle (v-) SNAREs with target (t-) membrane SNAREs forms the *trans*-SNARE complex that fuses the two opposing lipid bilayers, opens up a fusion pore, and culminates in cargo release from vesicles. This mechanism is critical for vesicular transport and is subjected to explicit control by several regulatory proteins that ensure cargo delivery to the right place at the right time[29,30]. Understanding the impact of these regulators on the *trans*-SNARE complex is challenging due to the astonishing speed of vesicle exocytosis[31]. Here, we reconstituted v-SNAREs into APEX2-MSP1D1 NDs and readily observed its association with t-SNAREs embedded in liposomes (Fig. 5B and Fig. S1), with an apparent affinity of 2.8 µM. Moreover, these *trans* interactions were facilitated by syt1, which is known to promote the assembly of the SNARE complex[17,32]. The labeling of t-SNARE was well observed at low v-SNARE concentrations and the affinity was increased by ~30-fold in the presence of syt1. Together, we conclude that the nanodisc-ID approach could also be useful to study transmembrane protein–protein interactions.

## Discussion

MPs play essential roles in numerous cellular signaling processes[1,2]. However, structural and functional characterizations of MPs present a formidable challenge due to the hydrophobic nature of lipids and low-affinity interactions. To overcome these difficulties, we have developed nanodisc-ID that leverages the

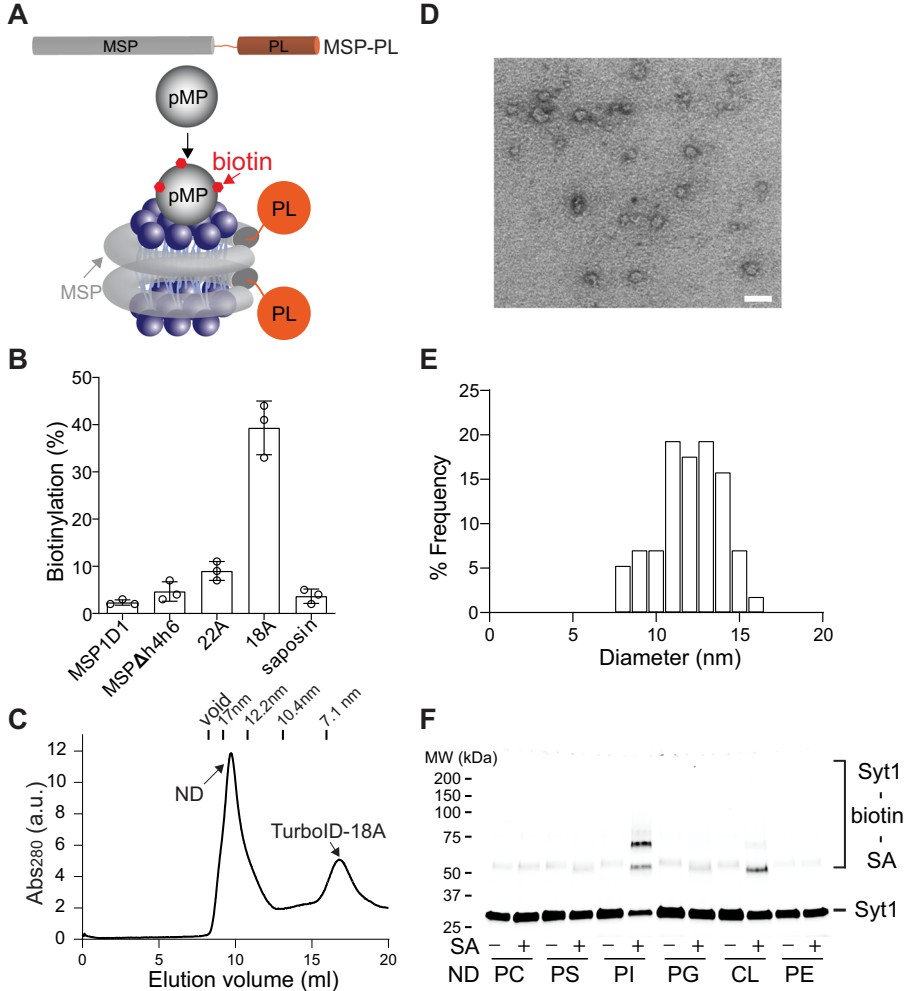

**Fig. 2 Developing nanodisc-ID to characterize protein–lipid interactions. A** Illustration of nanodisc-ID. Top, PL enzymes were fused to MSPs for developing the nanodisc-ID approach. Bottom, binding of pMPs to nanodiscs results in biotin labeling by PL enzymes. **B** The indicated MSPs were fused with TurboID and were then used to form nanodiscs containing PI lipids. PL efficiencies of syt1 by these nanodiscs were quantified. Data are shown as mean ± s.d., $n = 3$ independent experiments. **C** Representative size-exclusion chromatography (SEC) profile of TurboID-18A NDs containing PI lipids. **D** Representative negative stain image of the fractions corresponding to NDs from SEC in **C**. Scale bar: 20 nm. **E** Quantification of TurboID-18A ND diameters containing PI lipids from negative stain EM ($n = 57$). **F** The cytosolic domain of synaptotagmin-1 (syt1) was labeled with 5-IAF and incubated with TurboID-18A nanodiscs containing 10% of the indicated lipids and 90% of PC, followed by PL reactions. Samples, with or without the addition of SA, were subjected to SDS-PAGE and in-gel fluorescent imaging. PC 1,2-dioleoyl-sn-glycero-3-phosphocholine, PS 1,2-dioleoyl-sn-glycero-3-phospho-l-serine, PE 1-palmitoyl-2-oleoyl-sn-glycero-3-phosphoethanolamine, PI brain L-α-phosphatidylinositol-4,5-bisphosphate, CL *E.coli* cardiolipin.

power of PL with ND to detect membrane protein–lipid and protein–protein interactions. We expect that nanodisc-ID would greatly promote global mapping of protein–membrane interactions and systematic therapeutic discovery against cell-surface targets.

We showed that nanodisc-ID could enable versatile and robust characterizations of both peripheral and integral MPs. Empowered by the superior sensitivity of PL, nanodisc-ID could detect a range of membrane interactions with affinity from 10 nM to 100 μM (Figs. 3 and 5). Equilibrium titrations showed that the measured apparent binding affinities are similar to the reported values in previous studies, indicating that the fused PL enzymes did not perturb these interactions. In these experiments, the lipids incorporated in nanodiscs were chosen based on previous studies to benchmark the utility of nanodisc-ID[22–25]. With the continuous development of NDs[33,34], we believe that more complex membrane systems could be readily characterized. However, since it is not a direct measurement of binding affinities, results obtained using nanodisc-ID should be

further validated by other biophysical approaches, especially for unknown protein–membrane interactions.

In this study, we labeled MPs with fluorescence dyes for gel-imaging as it only required basic biochemical equipment and could confer rapid quantitative analysis. However, fluorescence labeling is not a necessity for the use of nanodisc-ID. In practice, any protein that interacts with the target in ND could be rapidly labeled with biotin, which is amenable to detection via many techniques such as mass spectrometry, fluorescence imaging, electron microscopy, and western blot[4]. Thus, nanodisc-ID is a label-free and versatile approach that could be implemented in a few hours in most biochemistry labs.

Since only the interacting proteins were biotinylated, nanodisc-ID could be employed to study protein–membrane interactions alone or in mixed samples. Here, we demonstrated the potential of nanodisc-ID to study competitive pMP-lipid interactions (Fig. S3). Moreover, we expect that synergistic protein–membrane interactions could also be interrogated using nanodisc-ID. This advantage would be highly useful to characterize signaling

pathways that involve a multitude of membrane proteins, such as the formation of immune synapses and the triggering of neuro-transmitter release[28,35]. In conjunction with high-throughput approaches, we anticipate that nanodisc-ID could also greatly facilitate the panning of antibody and peptide libraries against membrane targets.

Our study also demonstrated that PL could be used as a quantitative approach to determine membrane protein–lipid and protein–protein interactions. Even though PL could tag transient interactions, it is different from crosslinking approaches in the sense that the interacting molecules are not covalently linked together and are still in an equilibrium of association and dissociation. Thus, the labeling efficiency of the protein of interest

specifically depends on the binding of the target in nanodiscs. The derived apparent affinities are therefore very similar to the affinities of these interactions from previous studies (Fig. 3B).

One limitation of nanodisc-ID is the inability to characterize *cis* membrane interactions because PL enzymes would unspecifically label all the proteins in nanodiscs. Thus, we mainly focused on using nanodisc-ID for transmembrane interactions, in which the protein of interest is not co-reconstituted with the target in nanodiscs and would only be specifically labeled if bound to the target. Another problem of nanodisc-ID could arise due to the promiscuity of proximity labeling. This issue might cause false positives and would require proper control experiments for validation. Nevertheless, recent studies have also developed quantitative approaches and new PL substrates to mitigate this problem[4,36].

Finally, our study demonstrated an enormous potential to functionalize NDs. On this front, we are amazed by the fact that MSPs could be fused with PL enzymes and are still able to form NDs. In line with this discovery, MSPs were previously modified with functional groups for an array of applications in basic and translational research[10]. Combining with DNA origami, it is recently shown that MSPs could enable the construction of large NDs with diameters over 100 nm[37]. Together, we believe that MSPs could be further engineered to expand our toolkits for the characterization of membrane proteins.

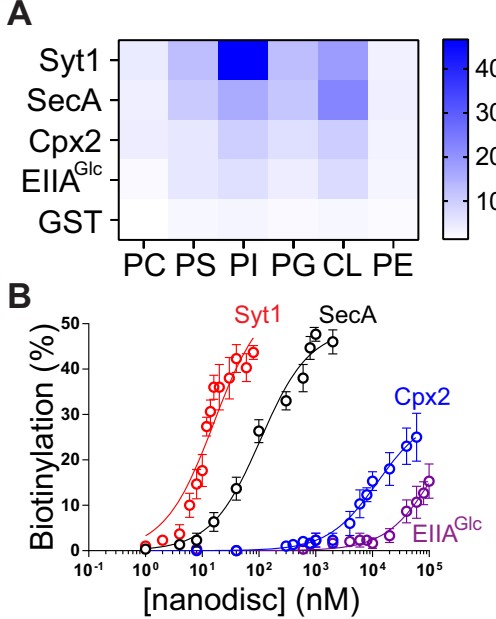

**Fig. 3 Using nanodisc-ID to profile and quantify pMP binding to lipids. A** Heat map of PL efficiencies of the indicated proteins by TurboID-18A nanodiscs containing the indicated lipids. **B** Equilibrium titration of TurboID-18A nanodiscs with peripheral proteins. The interactions of syt1 with PI ($K_d$ = 16 nM), SecA with CL ($K_d$ = 106 nM), cpx2 with PI ($K_d$ = 12 μM), and EIIA^Glc with CL ($K_d$ = 140 μM) were characterized by PL using TurboID-18A nanodiscs. PL efficiencies of the indicated proteins were then plotted against TurboID-18A nanodisc concentrations. Apparent binding affinities were determined by fitting to a one-site binding equation using GraphPad. Data are shown as mean ± s.d., n = 3 independent experiments.

## Methods

**Chemicals and reagents.** 1,2-dioleoyl-sn-glycero-3-[(N-(5-amino-1-carbox-ypentyl)iminodiacetic acid)succinyl] (DGS-NTA), 1,2-dioleoyl-sn-glycero-3-phos-phocholine (PC), 1,2-dioleoyl-sn-glycero-3-phospho-l-serine (PS), 1-palmitoyl-2-oleoyl-sn-glycero-3-phosphoethanolamine (PE), brain L-α-phosphatidylinositol-4,5-bisphosphate (PI), and *E.coli* cardiolipin (CL) were obtained from Avanti Polar Lipids. Nitrilotriacetic acid (Ni^2+-NTA)-chelating Sepharose and Superdex 200 increase 10/300 GL were purchased from GE Healthcare. All other chemicals were acquired from Sigma.

**Plasmids.** pTRC-APEX2, pET21a-BirA, and pET21a-TurboID were gifts from Dr. Alice Ting[13,15]. pET21a-BioID was generated by site-directed mutagenesis from pET21a-BirA. pGEX6p-1-PafA, and pGEX6p-1-BCCP-PupE were gifts from Dr. Min Zhuang[16]. pET28a-MSP1D1 was a gift from Dr. Steven Sligar[8]. TurboID-18A and APEX2-MSP1D1 were constructed into pET28a using the In-Fusion® HD Cloning Kit (Takara Bio USA).

**Proteins.** syt1, cpx2, SecA, MalFGK₂, MalE, EIIA^Glc, MSP1D1, and SNAREs were expressed in BL21 STAR™ (DE3) cells and purified using GSTrap or NTA-Ni^2+ columns[22,27,38–41]. To produce TurboID-18A and APEX-MSP, plasmids were transformed into BL21™ (DE3) STAR cells that were grown in LB supplemented with Km (50 mg/ml) to OD₆₀₀ ~0.5. Protein expression was induced with 0.2 mM IPTG at 16 °C overnight. Bacterial were harvested by centrifugation at 3700 rpm for 20 min, resuspended in Buffer A (50 mM Tris-HCl (pH 8),100 mM NaCl, 5% glycerol, 2 mM β-mercaptoethanol), and lysed using a Branson cell disrupter. Cell lysates were clarified by centrifugation at 10,000 rpm for 45 mins. The supernatants

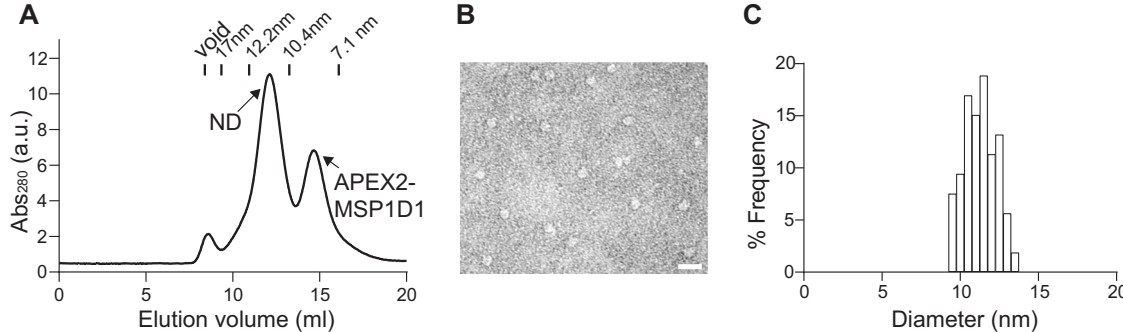

**Fig. 4 Characterization of APEX2-MSP1D1 nanodiscs. A** Reconstituted nanodiscs were separated by SEC using superdex 200 10/300. **B** Representative negative stain EM images of the fractions corresponding to NDs from SEC (**A**). Scale bar: 20 nm. **C** Quantification of ND sizes from negative stain EM images (n = 54).

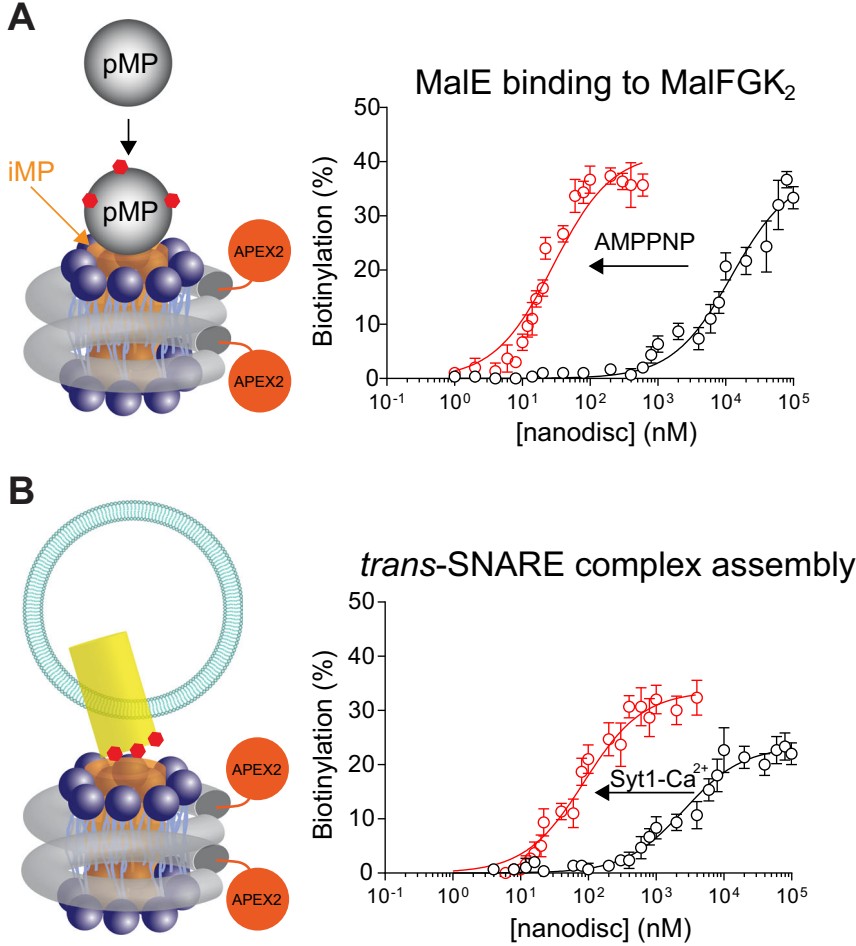

**Fig. 5 Using nanodisc-ID to characterize the interaction of membrane protein–protein interactions. A** Left, illustration of using APEX2-MSP1D1 NDs to study the association of pMP with integral membrane protein (iMP). Binding of pMP to iMP brings it to close distance with APEX2, which in turn labels pMP with biotin. Right, equilibrium titrations of APEX2-MSP1D1 NDs bearing MalFGK$_2$ with MalE in the presence ($K_d = 28$ nM) or absence ($K_d = 13$ μM) of AMPPNP. **B** Left, illustration of using APEX2-MSP1D1 NDs to study the *trans* interaction between two iMPs. Right, equilibrium titrations of APEX2-MSP1D1 NDs bearing v-SNAREs with liposomes harboring t-SNAREs. PL efficiencies of t-SNAREs were then plotted against v-SNARE nanodiscs concentrations with ($K_d = 86$ nM) or without ($K_d = 2.6$ μM) syt1-Ca$^{2+}$. Apparent binding affinities were determined by fitting to a one-site binding equation using GraphPad. Data are shown as mean ± s.d., $n = 3$ independent experiments.

were loaded onto a 1 ml NTA column (GE healthcare), followed by two times wash using buffer B (50 mM Tris-HCl (pH 8), 20 mM Imidazole, 400 mM NaCl, 5% glycerol, 2 mM β-mercaptoethanol). Proteins were eluted in buffer C (50 mM Tris-HCl (pH 8), 500 mM Imidazole, 400 mM NaCl, 5% glycerol, 2 mM β-mercaptoethanol), desalted in buffer A using PD MiDiTrap G-25 (GE Healthcare), and stored at −80 °C.

**Fluorescent labeling of proteins**. We labeled protein of interests using 5-IAF to allow quantification via in-gel fluorescence. Purified proteins were desalted using Zeba Spin columns (Thermo Fisher) in buffer D (50 mM Tris-HCl, pH 8, 100 mM NaCl, 5% glycerol) and labeled with a 3-fold excess of 5-IAF in the presence of TCEP (0.2 mM) at room temp for 3 h. We removed excessed dyes using Zeba Spin columns in buffer A.

**Nanodiscs**. For nanodiscs containing only lipids, MSPs were incubated with lipids at a ratio of 1:60. To prepare nanodiscs for profiling protein–lipid interactions, the indicated lipids were mixed with PC at a ratio of 1:9 in chloroform. Lipid mixtures were dried under a stream of nitrogen and resuspended in Buffer E (20 mM Tris-HCl (pH 8) and 1 mM DTT). For nanodiscs harboring iMPs, the indicated iMPs, MSPs, and PC lipids, were mixed at a ratio of 1:5:200 in buffer A containing 0.01% DDM. Detergents were slowly removed by gentle shaking with BioBeads (4 °C, overnight). Samples were purified by gel filtration using Superdex 200 10/300 (GE Healthcare) in buffer A and stored at −80 °C.

**Proximity labeling assay**. Proteins of interest (10 nM) were incubated with the indicated nanodiscs at room temp in PBS for 15 min. Samples were then subjected to proximity labeling by TurboID or APEX2 at room temperature (25 °C). We

performed TurboID-mediated labeling reactions in the presence of 0.5 mM biotin, 1 mM ATP and 5 mM MgCl$_2$ for 10 mins, whereas APEX2-mediated reactions were carried out with 0.5 mM H$_2$O$_2$ and 0.2 mM biotin-phenol for 1 min. Reactions were stopped by the addition of 0.1% SDS and then passed through Zeba Spin desalting columns. Samples were incubated with 5 μg of SA at room temp for 5 min and then subjected to SDS-PAGE and in-gel fluorescence imaging. PL efficiencies were calculated from the decreased intensity of the protein monomer bands before and after the addition of SA.

**Negative stain electron microscopy**. Formvar/carbon-coated copper grids (01754-F, Ted Pella, Inc.) were glow discharged (15 mA, 25 secs) using PELCO easiGlow$^{TM}$ (Ted Pella, Inc). NDs (20 μg/ml) were applied onto the grids for 30 secs, followed by staining with 0.75% uranyl formate for 1 min. Imaged were collected using a ThermoFisher Science Tecnai G2 TEM (100 kV) equipped with a Veleta CCD camera (Olympus). All TEM data were analyzed using Fiji to determine ND sizes.

**Other methods**. SDS-PAGE were performed using BioRad Mini-PROTEAN TGX precast protein gels (4–15%). ATPase assays were carried out using photocolorimetric method[27,38].

**Reporting summary**. Further information on research design is available in the Nature Research Reporting Summary linked to this article.

## Data availability
All original data will be made available by the corresponding authors upon reasonable request. Source data for Figs. 3 and 5 are in Supplementary Data 1.

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

## Acknowledgements

We thank Drs. Naomi Kamasawa and Debby Guerrero-Given from the imaging center at the Max Planck Florida Institute for Neuroscience for assist in EM. This work was made possible by support from the Scripps Research Institute and the NIH Director's New Innovator Award (DP2GM140920-01 to B.H.).

## Author contributions

B.H. conceived the project, performed the experiment, analyzed the data, and wrote the manuscript.

## Competing interests

The author declares no competing interests.
