## [Peer Review File · Communications Biology]

Reviewers' comments:

Reviewer #1 (Remarks to the Author):

The author made nanodisc coupled with proximity labeling enzymes. The author used the new nanodisc (nanodisc ID) to quantitatively analyze membrane proteins-protein interactions as well as membrane protein-lipid interactions using a gel electrophoresis readout. While the idea is interesting, there is a lack of important data to help evaluate the new approach.

Major concerns

A major concern is the lack of any data that demonstrate the ability of Turbo-ID-18A or APEX2-MSP1D1 to properly assemble nanodiscs. At Least size exclusion profiles and negative stain EM images should be shown. Also data regarding the homogeneity and the stability of the new Turbo-ID-18A or APEX2-MSP1D1 nanodiscs should be presented.

Minor concerns

- Figure 1B is confusing. There is no molecular weight ladder. Also, the top band can be seen without adding SA? Is this a dimeric Syt1?
- The gel images that are used to generate the heat map in figure 2A should be included in the supplementary section.
- The author labeled the proteins of interest with 5-IAF yet it is called label-free approach. This should be corrected/clarified.

Reviewer #2 (Remarks to the Author):

The paper by Huan Bao describes an interesting approach for studying membrane protein interactions within a nanodisc platform. This is an important area and new tools to study lipid and protein interactions are much needed in the field.

The brief nature of the report and the multiple samples tested do at times make the manuscript hard to read and at times more data would be informative. By having no clear subsections within the manuscript it is hard to read. It would be good to have more defined sections within the manuscript.

One particular concern that should be addressed is that if we have for example a membrane protein within a nanodisc and then labelled lipids which are not only free to move but also the labelling distance is 10-20nm then where does the selectivity come from? A nanodisc is typically only 10-20 nm in diameter so by placing both the target and tag within this system would this not create a lot of false positives? If for example a nanodisc was filled with different lipids it is not clear why you would get different labelling efficiencies as the distance within the nanodisc means even without a close interaction you may get good labelling efficiency. Even a brief transient interaction could produce a false positive. This can be a problem for any crosslinking experiment where even a short lived state can be trapped and populate over time to make it look more frequent than it is. I maybe missing something, but it would be good for this to be discussed in the manuscript. Could a difference in labelling simply be that there are differences in the efficiency of the incorporation of the protein within the nanodisc? It would be good to see data to show the level of protein in the nanodisc?

The labelling efficiency is ~5-50% in figure S1, could this be improved?

In figure 1 why is so much of the Syt1 protein unlabelled this is by far the dominant band. A Mw ladder would also be a welcome addition to the figure.

Is there a difference in labelling efficiency of the lipids themselves using different Ni²⁺ lipids?

Would it not be better to have a mixture of lipids in some of the experiments. The tight binding of an unlabelled lipid type to a protein may protect from labelling from a different lipid type and you could

carry out a competition assay.

It would be good to see more data on the quality of the protein that is going into the nanodisc, even just a simple gel would be good.

How do you decide on the lipids to incorporate back into the nanodisc, for more complex membrane systems this will be challenging and require an appropriately modified lipid?

Could the reporter become the news in that the addition of the tag increases the affinity of the lipid to the protein? Or vice versa the tag changes the characteristic such that it no longer binds?

In conclusion I think that this is an interesting idea and it could become a useful technology for a range of applications. However, given the briefness of the paper and the amount of data it is trying to convey I find it hard to follow the work and it raises a number of questions that should be addressed.

Reviewer #3 (Remarks to the Author):

This is a brief methods paper with quick outlines of many proof-of-principle experiments. No detailed description, no discussion, no explanation of methods and chosen conditions. The idea is good, but the paper reads like a brief outline. This should be augmented, putting in answers such as to - what if? why this label? It needs to be submitted to a methods journal such as BioTechniques.

Reviewer #1 (Remarks to the Author):

The author made nanodisc coupled with proximity labeling enzymes. The author used the new nanodisc (nanodisc ID) to quantitatively analyze membrane proteins-protein interactions as well as membrane protein-lipid interactions using a gel electrophoresis readout. While the idea is interesting, there is a lack of important data to help evaluate the new approach.

Response: We thank this reviewer's enthusiasm for our work. We have performed further experiments to analyze nanodiscs as detailed below. We hope that the reviewer finds the revised manuscript satisfactory.

Major concerns

1. A major concern is the lack of any data that demonstrate the ability of Turbo-ID-18A or APEX2-MSP1D1 to properly assemble nanodiscs. At Least size exclusion profiles and negative stain EM images should be shown. Also data regarding the homogeneity and the stability of the new Turbo-ID-18A or APEX2-MSP1D1 nanodiscs should be presented.

Response: All the nanodiscs used in this study were purified and characterized using size exclusion chromatography (SEC). We initially submitted this manuscript to Nat Methods as a Brief Communication, which was directly transferred to Communications Biology. Thus, we did not include those data, due to the length limitation of Nat Methods. We have now provided these data and also performed negative stain EM to characterize nanodiscs. The data confirmed the ability of TurboID-18A and APEX2-MSP1D1 to form monodisperse nanodiscs with diameters of 10-15 nm (new Fig. S2 and S7). In addition, these nanodiscs were stable at 4 °C for at least 24 hours (new Fig. S4). In the revised manuscript, we have included these data in the new Fig. S2, S4 and S7. We have made the following changes:

“Gratifyingly, fusion with a membrane scaffold peptide (MSP-18A) could still form homogenous NDs (Fig. S2B-D) and did not perturb the activity of TurboID for detecting the interaction between syt1 and lipids (Fig. 1, Figs. S2-4), yielding TurboID-18A.”

“We fused APEX2 with MSP1D1, and the resulting APEX2-MSP1D1 was able to form homogenous and stable NDs (Fig. S4 and S7).”

Negative stain electron microscopy

Formvar/carbon-coated copper grids (01754-F, Ted Pella, Inc.) were glow discharged (15 mA, 25 secs) using PELCO easiGlow™ (Ted Pella, Inc). NDs (20 µg/ml) were applied onto the grids for 30 secs, followed by staining with 0.75% uranyl formate for 1 minute. Images were collected using a ThermoFisher Science Tecnai G2 TEM (100 kV) equipped with a Veleta CCD camera (Olympus). All TEM data were analyzed using Fiji to determine ND sizes.”

Minor concerns

1- Figure 1B is confusing. There is no molecular weight ladder. Also, the top band can be seen without adding SA? Is this a dimeric Syt1?

Response: We apologize for the confusion. In the revised manuscript, we have re-done these experiments and added a molecular weight ladder in the new Fig. 1B.

Regarding the top band seen without adding SA, there might be several possibilities. It could be syt1 oligomers formed in the presence of lipids (Bello, et al., 2018 PNAS). Another possibility is that syt1 is labeled with multiple biotins that cause mobility shift on gel electrophoresis without the addition of SA. Finally, since we stopped PL reactions with SDS and then performed rapid buffer exchange to remove free biotin, a fraction of the protein aggregated during this process and showed up as oligomers on SDS-PAGE. Thus, we are not sure as to the exact reason for these top bands. Nevertheless, upon incubation with SA, higher molecular weight bands showed up, and the intensity of the syt1 monomer band significantly decreased when using nanodiscs containing negatively charged lipids such as PIP2. Thus, syt1 is biotinylated when bound to nanodiscs conjugated with TurboID. For data analysis, we therefore only quantified the decreased intensity of the syt1 monomer band after the addition of SA. To clarify this issue, we have made the following changes in the revised manuscript:

“PL efficiencies were calculated from the decreased intensity of the protein monomer bands before and after the addition of SA.”

2- The gel images that are used to generate the heat map in figure 2A should be included in the supplementary section.

Response: We have provided the images in the new Fig. S6 of the revised manuscript and made the following changes:

“Using TurboID-18A NDs prepared with different lipids, we could readily profile the lipid-binding specificity of several pMPs (Fig. 2A, Fig. S3 and S6)”.

3- The author labeled the proteins of interest with 5-IAF yet it is called label-free approach. This should be corrected/clarified.

Response: We indeed labeled the proteins of interest with 5-IAF to enable accurate and rapid quantification by in-gel fluorescence. However, the intensities of these protein bands could also be quantified by coomassie staining or western blot, and thus do not require fluorescence labeling. Moreover, since nanodisc-ID uses proximity labeling to biotinylate binding partners, these interactions could also be detected using mass spectrometry and electron microscopy. As a new lab started during the COVID-19 pandemic, we have not established all of these tools yet, but they are all feasible and label-free. We have clarified this issue in the revised manuscript:

“In this study, we labeled MPs with fluorescence dyes for gel-imaging as it only required basic biochemical equipment and could confer rapid quantitative analysis. However, fluorescence labeling is not a necessity for the use of nanodisc-ID. In practice, any protein that interacts with the target in ND could be rapidly labeled with biotin, which is amenable to detection via many techniques such as mass spectrometry, fluorescence imaging, electron microscopy, and western blot⁴. Thus, nanodisc-ID is a label-free and versatile approach that could be implemented in a few hours in most biochemistry labs”.

Reviewer #2 (Remarks to the Author):

The paper by Huan Bao describes an interesting approach for studying membrane protein interactions within a nanodisc platform. This is an important area and new tools to study lipid and protein interactions are much needed in the field.

Response: We are very grateful for the enthusiasm of this reviewer in our work.

The brief nature of the report and the multiple samples tested do at times make the manuscript hard to read and at times more data would be informative. By having no clear subsections within the manuscript it is hard to read. It would be good to have more defined sections within the manuscript.

Response: We initially submitted this manuscript to Nat Methods as a Brief Communication, which was directly transferred to Communications Biology. So, the manuscript has no subsections and thus is indeed hard to follow. We apologize for this inconvenience, and we have reformatted the manuscript as a formal article to elaborate on our work. In addition, we have carried out further experiments to explain the nanodisc-ID approach. We hope that the reviewer finds the revised manuscript satisfactory.

1. One particular concern that should be addressed is that if we have for example a membrane protein within a nanodisc and then labelled lipids which are not only free to move but also the labelling distance is 10-20nm then where does the selectivity come from? A nanodisc is typically only 10-20 nm in diameter so by placing both the target and tag within this system would this not create a lot of false positives? If for example a nanodisc was filled with different lipids it is not clear why you would get different labelling efficiencies as the distance within the nanodisc means even without a close interaction you may get good labelling efficiency. Even a brief transient interaction could produce a false positive. This can be a problem for any crosslinking experiment where even a short lived state can be trapped and populate over time to make it look more frequent than it is. I maybe missing something, but it would be good for this to be discussed in the manuscript. Could a difference in labelling simply be that there are differences in the efficiency of the incorporation of the protein within the nanodisc? It would be good to see data to show the level of protein in the nanodisc?

Response: We apologize for this confusion on lipid selectivity. First, we would like to clarify that we did not study the interaction of lipids with membrane proteins co-reconstituted in nanodiscs. For characterization of protein-lipid interactions using nanodisc-ID, we focused on periphery membrane proteins (pMPs) that were not incorporated in nanodiscs. In these experiments (Figs. 1 and 2), nanodiscs encased different lipids, and pMPs were added in solution (Fig. 1A). Thus, the selectivity of labeling efficiencies depends on the binding of pMPs to the specific lipids in nanodiscs conjugated with proximity labeling (PL) enzymes.

Second, we thought that the confusion might arise from our initial screen using Ni²⁺-NTA lipids to immobilize PL enzymes (Fig. S1). We found that pMPs could bind Ni²⁺-NTA lipids even though they did not contain a Histag, and as a consequence, a fraction of pMPs were immobilized together with PL enzymes on nanodiscs in this scenario, resulting in low lipid

selectivity (Fig. S1). To resolve this problem, we genetically fused PL enzymes with MSPs in all the following experiments as described in the main figures, thereby bypassing the use of Ni²⁺-NTA lipids in the nanodisc-ID approach. In doing so, the lipid selectivity is greatly improved (Figs. 1 and 2), as it lies in the specific interaction of pMPs with the lipids in nanodiscs.

To further clarify this issue, we pointed out that nanodisc-ID is not suitable for the characterization of *cis* membrane interactions. In addition, we included a new figure showing a gel of pMPs and nanodiscs used in this study (Fig. S3). We have also revised the illustrations in Fig. 1A and Fig. S1A to clarify the different NDs used in the initial screens. In the revised manuscript, we have made the following changes:

“Using TurboID-18A NDs prepared with different lipids, we could readily profile the lipid-binding specificity of several pMPs (Fig. 2A, Fig. S3 and S6)”

“One limitation of nanodisc-ID is the inability to characterize *cis* membrane interactions because PL enzymes would unspecifically label all the proteins in nanodiscs. Thus, we mainly focused on using nanodisc-ID for *trans* membrane interactions, in which the protein of interest is not co-reconstituted with the target in nanodiscs and would only be specifically labeled if bound to the target”.

2. The labelling efficiency is ~5-50% in figure S1, could this be improved?

Response: Yes, we could improve the labeling efficiency. We tried extensively on this front and found that PL labeling efficiencies were mainly limited by the affinities and kinetics of pMPs binding to lipids. During PL experiments, we observed that PL enzymes were not stable and could also label themselves. We believe that these two problems cause these enzymes to quickly lose their PL activities. As such, weak lipid-binding pMPs will have lower labeling efficiency. In contrast, PL efficiency could reach 100% if the binding of pMPs to lipids could be improved. For example, we found that increasing the ratio of Ni²⁺-NTA lipids to 50% in nanodiscs could result in 100% labeling of syt1 (new Fig. S1D). We believe that this is because syt1 binds to Ni²⁺-NTA lipids much better and faster than any other lipids. However, the use of Ni²⁺-NTA lipid makes it no longer possible to characterize the specificity of syt1-lipid interactions, and we moved on to find new ways to bypass the use of Ni²⁺-NTA lipids for developing the nanodisc-ID approach. Nevertheless, the data obtained from the use of Ni²⁺-NTA lipids indicated that it is feasible to obtain higher labeling efficiency if pMP-lipid interactions have faster kinetics and higher affinities.

In the revised manuscript, we have included this result in the new Fig. S1B-D, and we have made the following change:

“We found that the labeling efficiency of syt1 is correlated with the percentage of Ni²⁺-NTA conjugated lipids, suggesting that PL labeling efficiencies reflect the affinities of protein-lipid interactions (Fig. S1B and D)”.

3. In figure 1 why is so much of the Syt1 protein unlabelled this is by far the dominant band. A Mw ladder would also be a welcome addition to the figure.

Response: Binding of syt1 to these nanodiscs determines the labeling efficiency. As mentioned above, if syt1 binding to nanodiscs was mediated by Ni²⁺-NTA lipids, its labeling efficiency could reach 100%. So, the fact that syt1 labeling by nanodiscs in Fig. 1 is much lower, suggests that the association of these lipids with syt1 is quite dynamic and much weaker than Ni²⁺-NTA lipids. In addition, it also suggests that the nanodisc-ID approach could be used to determine the affinity of protein-lipid interactions.

To clarify the different bands on the SDS-PAGE, we have re-done these experiments and added a MW ladder on the gel in the new Fig. 1B of the revised manuscript.

4. Is there a difference in labelling efficiency of the lipids themselves using different Ni²⁺ lipids?

Response: Regarding the labeling of lipids themselves, it is not known that TurboID and APEX2 could react with lipids. Thus, we expect that lipids were not labeled in our experiments. However, we are also keen to revamp our approach for lipid characterizations. On this front, we could readily carry out these studies in the future when new versions of substrates and PL enzymes that could label lipids become available.

Moreover, since we found that the lipid selectivity is low using Ni²⁺-NTA lipids, we did not pursue this direction and moved on to genetically fuse PL enzyme with MSPs. Using these new MSPs (e.g., TurboID-18A), PL efficiencies of pMPs become lipid selective (Fig. 2A), and thus we stopped using Ni²⁺-NTA lipids in all the main figures. During the revision, we did further test the possibility of using nanodiscs harboring Ni²⁺-NTA and different lipids (Fig. S1D). As expected, we did not observe lipid specificity in PL labeling of syt1. So, we thought that it is better not to use Ni²⁺-NTA lipids for developing the nanodisc-ID approach.

We have included these data in the new Fig. S1D of the revised manuscript and made the following changes:

“Furthermore, the lipid specificity of syt1 was not captured (Fig. S1B and D), and other low-affinity protein-lipid interactions could not be detected (Fig. S1C). We suspected that these data were due to the use of Ni²⁺-NTA conjugated lipids; proteins such as syt1 could bind cationic ions (e.g., Ni²⁺) and the lipid anchored TurboID could constrain the surface area of NDs for pMP association. Nevertheless, we found that the labeling efficiency of syt1 is correlated with the percentage of Ni²⁺-NTA conjugated lipids, suggesting that PL labeling efficiencies reflect the affinities of protein-lipid interactions (Fig. S1B and D)”.

5. Would it not be better to have a mixture of lipids in some of the experiments. The tight binding of an unlabelled lipid type to a protein may protect from labelling from a different lipid type and you could carry out a competition assay.

Response: We thank the reviewer for this brilliant idea. Since we found that the labeling efficiency in nanodisc-ID is directly related to the affinity of pMPs for lipids, competition could occur when two different nanodiscs were used in these assays. This could further demonstrate the lipid specificity and showcase more potential applications of the nanodisc-ID approach. To test this idea, we have performed a competition experiment of syt1 binding to TurboID-18A nanodiscs bearing PS and regular MSP1D1 nanodiscs containing PIP2 (new Fig. S5). Indeed, PIP2 could compete with PS to bind syt1 and cause decreased labeling efficiencies of syt1 by

TurboID-18A nanodiscs. Therefore, nanodisc-ID could be used to study the competitive binding of lipids to pMPs.

We have included these data in the revised manuscript and made the following changes:

“In addition, the labeling efficiencies of syt1 by TurboID-18A NDs are consistent with its lipid specificity (Fig. 2A). As such, syt1 labeling by TurboID-18A NDs containing PS lipids was inhibited by regular MSP1D1 nanodiscs containing PI, but not PC lipids (Fig. S5)”

“Since only the interacting proteins were biotinylated, nanodisc-ID could be employed to study protein-membrane interactions alone or in mixed samples. Here, we demonstrated the potential of nanodisc-ID to study competitive pMP-lipid interactions (Fig. S5). Moreover, we expect that synergistic protein-membrane interactions could also be interrogated using nanodisc-ID. This advantage would be highly useful to characterize signaling pathways that involve a multitude of regulatory proteins, such as the formation of immune synapses and the triggering of neurotransmitter release^{28, 33}. In conjunction with high-throughput approaches, we anticipate that nanodisc-ID could also greatly facilitate the panning of antibody and peptide libraries against membrane targets”.

6. It would be good to see more data on the quality of the protein that is going into the nanodisc, even just a simple gel would be good.

Response: In the revised manuscript, we have provided a new Fig. S3, showing a gel for the proteins and nanodiscs used in this work.

7. How do you decide on the lipids to incorporate back into the nanodisc, for more complex membrane systems this will be challenging and require an appropriately modified lipid?

Response: The membrane protein-lipid and protein-protein interactions characterized in this work have been previously well studied. Thus, we chose lipids based on the literature to establish the feasibility and utility of the nanodisc-ID approach. We agree that other lipids need to be included for more complex membrane systems. This should not be too much of an issue for nanodiscs, as recent processes in nanodisc have significantly increased its ability to incorporate a diverse array of membrane proteins and lipids (Nasr, 2020 Curr Opin Struct Biol; Sligar and Denisov, 2020 Protein Sci). In the revised manuscript, we have included this point in the discussion:

“In these experiments, the lipids incorporated in nanodiscs were chosen based on previous studies to benchmark the utility of nanodisc-ID²²⁻²⁵. With the continuous development of NDs^{33, 34}, we believe that more complex membrane systems could be readily characterized using nanodisc-ID.”

8. Could the reporter become the news in that the addition of the tag increases the affinity of the lipid to the protein? Or vice versa the tag changes the characteristic such that it no longer binds?

Response: We agree that this issue could be a concern. Using equilibrium titrations (Fig. 2B and 3), we found that the affinities of several pMPs for lipids and membrane targets were consistent with previous studies. So, we think the effect is relatively small. However, nanodisc-ID is not a direct measurement of binding affinities and thus the resulting data should also be validated using other biophysical approaches, especially for unknown protein-membrane interactions. In the revised manuscript, we have further discussed this issue:

“Equilibrium titrations showed that the measured apparent binding affinities are similar to the reported values in previous studies, indicating that the fused PL enzymes did not perturb these interactions.”

“However, since nanodiscs-ID is not a direct measurement of binding affinities, results obtained using nanodisc-ID should be further validated by other biophysical approaches, especially for unknown protein-membrane interactions.”

In conclusion I think that this is an interesting idea and it could become a useful technology for a range of applications. However, given the briefness of the paper and the amount of data it is trying to convey I find it hard to follow the work and it raises a number of questions that should be addressed.

Response: We are very grateful for the enthusiasm from this reviewer and have further elaborated on our work in the revised manuscript as detailed above. We hope that the revised manuscript has addressed these problems.

Reviewer #3 (Remarks to the Author):

This is a brief methods paper with quick outlines of many proof-of-principle experiments. No detailed description, no discussion, no explanation of methods and chosen conditions. The idea is good, but the paper reads like a brief outline. This should be augmented, putting in answers such as to - what if? why this label? It needs to be submitted to a methods journal such as BioTechniques.

Response: We would like to thank the reviewer’s enthusiasm for the idea in our work, and we apologize for the short format. This is because we initially submitted this manuscript to Nat Methods as a Brief Communication, which was directly transferred to Communications Biology. So, the original manuscript has no subsections. In the revised manuscript, we have reformatted it as a formal article and have further explained our work. Moreover, we have carried out additional experiments to evaluate the nanodisc-ID approach. We hope that the reviewer finds these revisions satisfactory.

REVIEWERS' COMMENTS:

Reviewer #2 (Remarks to the Author):

The paper by Huan Bao has been revised and this has had a positive impact on the readability. By adapting this from the Nat. Method short communication format the manuscript now reads better and the data are better displayed. There could be a better balance in the figures, there are currently 3 main figures and 8 supplemental. Can some of these figures be moved into the main text, for example Fig S1, or S2 and S7 they back up the data presented? This could be a powerful approach and it would be interesting to see how this technology is picked up by the wider community. The manuscript is still very concise which can makes it difficult to follow in some sections. There are ~90 lines of text in the results and ~60 in the discussion and this covers 10 figures of data. As this is a new technique a good description for those not in the direct field will help in widening the impact and increasing the number of users of the technology.

Minor points

- The scale bar in figure S7 is not defined.
- The figure legend for 1B could be extended to define PC, PS, etc... to allow the figure to stand alone.

Reviewer #3 (Remarks to the Author):

This revised manuscript describes a new application of nanodisc technology, which is based on various methods of incorporation of several proximity labeled (PL) enzymes, with the best results achieved with APEX2 and TurboID. The first approach was based on attaching PL enzymes to Ni-NTA functionalized lipids via His-tag. However, some proteins bind to the Ni-NTA groups and thus generate high level of background error. Importantly, it was demonstrated that direct fusion of PL enzymes to the scaffold protein MSP or with short 18-peptides does not eliminate the ability of MSP and amphipathic peptides to make nanodiscs. This discovery opens the door to the label-free studies of protein-protein interactions mediated by the membrane surface of nanodiscs (with APEX2), as well as binding of various proteins to the nanodisc lipids (using TurboID). Specific interactions of synaptotagmin-1 with several lipids have been evaluated using this method, and the known PS specificity of this protein was confirmed, indicating the viability of this approach. Protein-protein interactions on nanodiscs have been probed using maltose transporter incorporated in nanodiscs and the SNARE complex formed between its constituent proteins v-SNARE in nanodiscs and t-SNARE in liposomes. All these experiments are well designed and clearly described, the manuscript provides enough experimental results to prove that this conceptual approach is feasible, although maybe not the most concise presentation. The paper can be published after several minor corrections listed below

The list of references should be formatted in a uniform manner. This is sloppy.
ref. 34 is incomplete.

Figure 2, legend, 1.2 - should this be "peripheral proteins"?
Figure 2 - GST? The full name is never introduced in the text.

Reviewer #2 (Remarks to the Author):

The paper by Huan Bao has been revised and this has had a positive impact on the readability. By adapting this from the Nat. Method short communication format the manuscript now reads better and the data are better displayed. There could be a better balance in the figures, there are currently 3 main figures and 8 supplemental. Can some of these figures be moved into the main text, for example Fig S1, or S2 and S7 they back up the data presented? This could be a powerful approach and it would be interesting to see how this technology is picked up by the wider community. The manuscript is still very concise which can makes it difficult to follow in some sections. There are ~90 lines of text in the results and ~60 in the discussion and this covers 10 figures of data. As this is a new technique a good description for those not in the direct field will help in widening the impact and increasing the number of users of the technology.

Response: We would like to thank this reviewer for the constructive suggestions that have led to a much-improved manuscript. Also, we feel very encouraged and inspired by the enthusiasm of the reviewer for our work. In this re-revised manuscript, we have included Fig S1, S2 and S7 as main figures (new Fig 1, 2 and 4) and have further explained our approach in the text.

We have made the following changes:

“Gratifyingly, fusion of TurboID with a membrane scaffold peptide (MSP-18A), yielding TurboID-18A, could still form homogenous NDs with a diameter of ~10-15 nm (Fig. 2B-D and Fig. S1), as characterized using size exclusion chromatography (SEC) and negative stain EM”

“In addition, the labeling efficiencies of syt1 by TurboID-18A NDs were consistent with its lipid specificity (Fig. 2F and 3A). As such, the highest labeling of syt1 occurred in the presence of PI lipids, and syt1 labeling by TurboID-18A NDs containing PS lipids was inhibited by regular MSP1D1 nanodiscs containing PI, but not PC lipids (Fig. S3). These data suggested that the nanodisc-ID approach could determine the specificity of protein-membrane interactions *in vitro*, even though proximity labeling tends to capture unspecific and transient binding events *in vivo*.”

“Interestingly, we noticed that syt1 and SecA were much better biotinylated by TurboID-18A NDs than cpx and EIIA^{Glc}, indicating that the labeling efficiencies were dependent on the affinities of pMPs for lipids (Fig. 3A).”

“We fused APEX2 with MSP1D1, and the resulting APEX2-MSP1D1 was able to form homogenous and stable NDs with a diameter of ~10-12 nm, as shown by SEC and negative stain EM (Fig. 4 and S2). Moreover, we could functionally reconstitute MalFGK₂ in APEX2-MSP1D1 NDs (Fig. S1 and S5), in which the ATPase activity of MalFGK₂ was responding to the addition of MalE and maltose.”

“we reconstituted v-SNAREs into APEX2-MSP1D1 NDs and readily observed its association with t-SNAREs embedded in liposomes (Fig. 5B and Fig. S1), with an apparent affinity of 2.8 μM.”

“Our study also demonstrated that PL could be used as a quantitative approach to determine membrane protein-lipid and protein-protein interactions. Even though PL could tag transient interactions, it is different from crosslinking approaches in the sense that the interacting molecules are not covalently linked together and are still in an equilibrium of association and dissociation. Thus, the labeling efficiency of the protein of interest specifically depends on the

binding of the target in nanodiscs. The derived apparent affinities are therefore very similar to the affinities of these interactions from previous studies (Fig. 3B).”

Minor points

-The scale bar in figure S7 is not defined.

-The figure legend for 1B could be extended to define PC, PS, etc... to allow the figure to stand alone.

Response: We have defined the scale bar in figure S7 (now Fig. 4) and the lipids in the figure legend of 1B (now Fig. 2F).

Reviewer #3 (Remarks to the Author):

This revised manuscript describes a new application of nanodisc technology, which is based on various methods of incorporation of several proximity labeled (PL) enzymes, with the best results achieved with APEX2 and TurboID. The first approach was based on attaching PL enzymes to Ni-NTA functionalized lipids via His-tag. However, some proteins bind to the Ni-NTA groups and thus generate high level of background error. Importantly, it was demonstrated that direct fusion of PL enzymes to the scaffold protein MSP or with short 18-peptides does not eliminate the ability of MSP and amphipathic peptides to make nanodiscs. This discovery opens the door to the label-free studies of protein-protein interactions mediated by the membrane surface of nanodiscs (with APEX2), as well as binding of various proteins to the nanodisc lipids (using TurboID). Specific interactions of synaptotagmin-1 with several lipids have been evaluated using this method, and the known PS specificity of this protein was confirmed, indicating the viability of this approach. Protein-protein interactions on nanodiscs have been probed using maltose transporter incorporated in nanodiscs and the SNARE complex formed between it's constituent proteins v-SNARE in nanodiscs and t-SNARE in liposomes. All these experiments are well designed and clearly described, the manuscript provides enough experimental results to prove that this conceptual approach is feasible, although maybe not the most concise presentation. The paper can be published after several minor corrections listed below.

Response: We are very grateful for the enthusiasm from this reviewer and would also like to thank the reviewer for the insightful critiques. In the re-revised manuscript, we have corrected the following problems as detailed below.

The list of references should be formatted in a uniform manner. This is sloppy. ref. 34 is incomplete.

Response: We apologized for this mistake. The reference list has now been carefully formatted.

Figure 2, legend, l.2 - should this be "peripheral proteins"?

Response: Yes, we have corrected it into 'peripheral proteins'.

Figure 2 - GST? The full name is never introduced in the text.

Response: We have now introduced the full name of GST (Glutathione-S-transferases) in the manuscript. We made the following changes:

“In control experiments, GST (Glutathione-S-transferase) that does not interact with lipids, was not labeled by TurboID-18A NDs at all conditions (Fig. 3 and Fig. S4).”